# Geometric Midpoint Algorithm for Device-Free Localization in Low-Density Wireless Sensor Networks

Chao Sun [1], Biao Zhou [2,3], Shangyi Yang [1] and Youngok Kim [1,*]

1 Electronic Engineering Department, Kwangwoon University, Seoul 01897, Korea; sunchao2601@gmail.com (C.S.); shangyigoo@gmail.com (S.Y.)
2 School of Internet of Things Engineering, Jiangnan University, Wuxi 214122, China; zhoubiao@jiangnan.edu.cn
3 Key Laboratory of Dynamic Cognitive System of Electromagnetic Spectrum Space, Ministry of Industry and Information Technology, Nanjing University of Aeronautics and Astronautics, Nanjing 211106, China
* Correspondence: kimyoungok@kw.ac.kr

**Abstract:** Device-free localization (DFL) is a technique used to track a target transporting no electronic devices. Radiofrequency (RF) tomography based DFL technology in wireless sensor networks has been a popular research topic in recent years. Typically, high-tracking accuracy requires a high-density wireless network which limits its application in some resource-limited scenarios. To solve this problem, a geometric midpoint (GM) algorithm based on the computations of simple geometric objects is proposed to realize effective tracking of moving targets in low-density wireless networks. First, we proposed a signal processing method for raw RSS signals collected from wireless links that can detect the fluctuations caused by a moving target effectively. Second, a geometric midpoint algorithm is proposed to estimate the location of the target. Finally, simulations and experiments were performed to validate the proposed scheme. The experimental results show that the proposed GM algorithm outperforms the geometric filter algorithm, which is a state-of-the-art DFL method that yields tracking root-mean-square errors up to 0.86 m and improvements in tracking accuracy up to 67.66%.

**Keywords:** device-free localization; moving target; radio frequency tomography; received-signal strength; wireless sensor networks

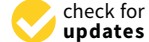



## 1. Introduction

Device-free localization (DFL) is a promising technology in wireless sensor networks (WSNs) that focuses on the detection of the position information and other moving status information of humans who do not carry any tractable electronic devices with them. Compared with the conventional active localization and tracking schemes that demand the target to carry devices, such as smartphones, radiofrequency identification (RFID) tags, or others, the DFL technology has unique superiority in various application scenarios, such as security safeguard systems of smart homes/buildings/factories, battle field surveillance systems, patient and elderly monitoring systems in hospitals or geriatric homes, wherein the targets rarely carry electronic devices which interact with monitoring system [1–7]. Limited by the disadvantage of being sensitive to illumination and by the poor performance of penetrating through walls and other nonmetallic obstacles, infrared sensors can only be used for restricted tasks [8]. Cameras can also be used in restricted scenarios because of privacy protection and legal limitations [9–12]. UWB technology has been known for its excellent detection accuracy in DFL, but its high cost is a limitation that renders it impractical [13,14]. Hence, WSN-based DFL technology is a better solution for contactless sensing and positioning.

Patwari et al. used the received-signal strength (RSS) variations relative to a vacant scene as observation information and modeled the DFL as a radio-tomography-imaging

problem to reconstruct the location [15–17]. Talampas et al. introduced a low-computational geometric filter (GF) algorithm to achieve accurate tracking [18,19]. The disadvantage of the above methods is that they make each wireless node of the high-density WSN, with the density of network approximately 1.25 link/m$^2$, broadcast signals in turn, which increases the energy consumption, complexity, and radio interference. Wang et al. introduced a lightweight robust Bayesian grid approach (BGA) to achieve robust location estimation [20]. Yang et al. developed a real-time radio-tomography-based DFL system with a compressed sensing algorithm which significantly reduced the number of RSS measurements. However, its high-computational complexity limited its application in scenarios with limited memory and computational resources [21].

Generally, the targets in RF tomography based DFL are not cooperative with the monitoring system. To realize high-tracking target accuracy, a high density of wireless links is necessary owing to the limited sensing area of a single wireless link. However, a high density of wireless links increases data redundancy, energy consumption, computational complexity, and signal interference from multipath effects, especially, in indoor environments. To solve the conflict between the demand for high accuracy and the problems mentioned above, we focus on the geometry of the intersection point among the links in the wireless network and propose a geometric method.

In this study, a geometric midpoint (GM) algorithm was proposed. The middle point of the segments formed by the intersection points among the links in the wireless network is treated as the location estimate in the GM algorithm. The network initialization for building segment information on each link should be completed before tracking the moving target. The proposed algorithm calculates the current location estimate based on the previous location estimate, including the determination of the segments on the triggered links and the calculation of the distance-based weight, which can effectively reduce the error caused by the false-positive detection. The location was estimated by the weighted mean of the middle points of the nearest segments to the previous location estimate of the triggered links. The weight was based on the distance from the middle point of the segment to the previous location estimate. The performance of the proposed scheme was evaluated using simulations and actual experiments. The results demonstrate that the proposed GM algorithm is an accurate and robust method for device-free localization.

The main contributions of this paper are summarized as follows:

- We propose a GM algorithm that can achieve accurate localization when the density of wireless network is 0.64 link/m$^2$, much lower than the network density 1.25 link/m$^2$ in conventional GF scheme [10]. The proposed algorithm uses the distance-weighted geometric midpoint of the segments with endpoints, which are the intersection points formed by the intersection of the links in the wireless network to estimate the position of the target. The algorithm can be applied to resource-limited scenarios
- We proposed an effective wireless link detection method that can detect the fluctuations caused by targets correctly, especially in indoor environments wherein the multipath effect is severe. As this is the first step of the proposed GM scheme, it makes important contributions to the final system tracking accuracy

The remainder of this paper is organized as follows. Section 2 reviews the related work about the state-of-the-art RF tomography-based device free tracking and localization schemes. Section 3 discusses the details of the proposed GM algorithm. In Section 4, the performance analysis and experimental results are presented. The concluding remarks are presented in Section 5.

## 2. Related Works

The RF tomography-based device-free tracking and localization has been attracting more and more researchers' concern recent several years. As the early researcher in this field, J. Wilson, and N. Patwari proposed a radio tomography imaging (RTI) method to solve the device-free target tracking problem [15–17]. They proposed a linear model to obtain the images of moving target by using of received signal strength (RSS) data. A noise

model based on real measurements is also provided. The mean-square error bound of RF tomography image is proposed to calculate the accuracy of RTI localization system [15]. An RSS variance related statistical model is introduced and is used to estimate the target motion images. Kalman filter is applied to recursively track the coordinates of the moving targets from the motion images [16]. Except tracking the dynamic moving targets, researchers also proposed measurement-based statistical model to tracking the moving and stationary targets in wireless network [17].

In addition to the imaging-based localization method, geometric based method with the advantage of low computation also become a new trend. M. C. R. Talampas et al. proposed a geometric filter (GF) algorithm to solve the device-free localization problem in the resource-restrained scenarios [18], which is as the conventional scheme compared with our proposed scheme. In GF scheme, accurate tracking is realized by using only operations on simple geometric objects, such as treating the intersection points of shadowed links as probable target locations. A circular prior region is defined and used to remove the outlying links and points. The location estimate is generated by the weighted mean of remaining points inside the circular prior region. The weights are based on the amount of shadowing experienced by the links and the distances of the intersection point to the prior location estimate. The updated multichannel geometric filter (MCGF) is proposed by utilizing the line-of-sight-link (LOSL) fade level and RSS variance on different frequency channels to detect the target triggering wireless links [19]. The Bayesian grid approach (BGA) was proposed by Wang et al. to solve the DFL problem using only lightweight operations on shadowing effect maps and employing prior and constraint information to realize a location estimate [20]. Recently, a channel impulse response (CIR) based method was proposed by Ninnemann et al. to locate the device-free targets combined with heatmap generated by radar imaging [22].

Recent two years, more advanced techniques are applied in device-free localization, such as machine learning and deep learning, etc. Zhang et al. provided a comprehensive survey of the state-of-the-art research on wireless sensing for human detection with a focus on wireless sensing systems (WSSs). A general structure of the deep learning (DL)-based WSS is introduced in detail for hitherto unexpected applications and future wireless sensing scenarios [23]. Zhao et al. formulated the DFL problem to an image classification problem and designed a three-layer convolutional autoencoder (CAE) neural network to perform unsupervised feature extraction from raw signals followed by supervised fine-tuning for classification [24]. Alberto et al. proposed a novel recurrent neural network (RNN) model-based adaptive indoor tracking framework combined with generative adversarial networks (GAN) to solve the accurate tracking and positioning problem in indoor environment [25]. Ma et al. designed a deep neural network to recognize the human gestures which can learn discriminative deep features and learn a transferrable similarity evaluation ability from the training set, and apply the learned knowledge to the new testing conditions [26]. Zhou et al. proposed a one-dimensional convolutional neural network (1D CNN) based method which exploits domain adaptation (DA) and semantic alignment (SA) to reduce the work on labor-intensive and time-consuming recalibrating in device-free WiFi localization [27]. Wang et al. designed an adversarial network (GAN) based mmWave FMCW system which realized high accuracy under a small training sample in device-free human gesture recognition [28]. Yan et al. proposed a novel decoupled convolutional neural network (CNN) based device-free activity detection and position estimation scheme using channel state information (CSI) in WiFi environment, which realized competitive performance in feature extraction compared with the state-of-the-art methods [29].

What's more, some conventional techniques in wireless communication field have been also updated and applied in device-free localization. Wang et al. proposed three novel multiplexing mechanisms–angle division multiplexing sensing, range division multiplexing sensing, and source division multiplexing sensing to realize multi-target device-free wireless sensing (DFWS) simultaneously [30]. Kaltiokallio et al. introduced a novel Bayesian filter which augments the measurement model of a Bayesian filter with position

estimates form an imaging approach. The filter nearly reach the posterior Cramer-Rao bound and is superior with respect to imaging approaches in terms of localization accuracy [31]. Zhao et al. modeled and implemented an accurate and easy-to-deploy system for indoor localization by combining with WiFi sensing technology and computer vision technology. This system enhances indoor localization with multimodal sensing vis two images, IMU sensors reading and CSI of WiFi signal, which realized accurate tracking accuracy [32]. Guo et al. proposed a new algorithm under the compressive sensing (CS) framework to track the time-varying target gestures in device-free localization (DFL) scheme [33]. Compared with the new techniques in recent years, our proposed method is a light weighed solution for low density wireless sensor network. The biggest advantages of our proposed method are the reliable link triggering strategy and simple localization principle, which makes it can be used in resource-limited scenarios and realize acceptable tracking accuracy.

## 3. Proposed GM Algorithm

The proposed GM algorithm scheme is presented in this section. Before starting, it is necessary that we should briefly introduce the conventional state-of-the-art GF scheme. In GF scheme, link filter (LF) and point filter (PF) are proposed to use the prior location estimate to build a circular prior region to: (1) remove outlier links; (2) remove improbable target locations; (3) assign distance-dependent weights to probable target locations ensuring robust tracking performance. The entire GM algorithm can be divided into several sections: detection the triggered links, intersection point algorithm, construction of link segments, computation of the midpoint of the segment and distance-based weights, and generation of the location estimate. All these aspects are introduced in the succeeding sections.

### 3.1. Detection of the Triggered Links

RF tomography based DFL is a technique used for the detection of a target carrying no electronic devices by using of the fluctuation of wireless signals of the links among sensors in a monitored area covered by a group of wireless sensors, as shown in Figure 1. Only a part of the wireless links near by the position of the target is triggered by the target, while all other links are not triggered. To enlarge the difference between the baseline and measured RSS caused by the appearance of the target in the vicinity of the wireless links and decrease the influence of environmental noise and sensor errors, we propose a robust detection method to detect triggered links that use a window to slide over the RSS data to capture the difference parts.

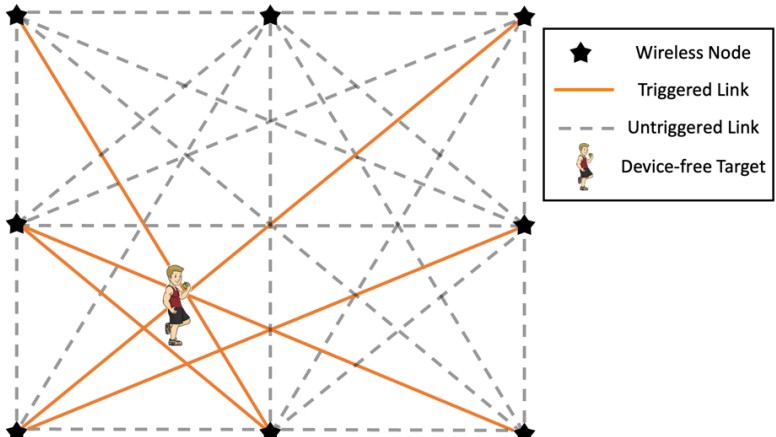

**Figure 1.** Device-free localization (DFL) system made up of eight nodes.

By applying a sliding window filter with a rectangular window function of length $c + 1$ $(c \ll Z)$ on the RSS measurement sequence $R_i = \{R_i(z) | z = 0, \ldots, Z - 1\}$ of the $i$-th

link $L_i$, where $Z$ is the measurement data length, we can obtain the subset of $R_i$ which is defined by,

$$R_i^n = \{R_i(z)|z = n, \ldots, n+c\}, \tag{1}$$

where $n$ is the start index of the rectangular window function which ranges from 0 to $Z - c - 1$.

The variance of the RSS subset $R_i^n$ can be calculated as,

$$V_i^n = \frac{1}{c+1} \sum_{z=n}^{n+c} (R_i(z) - \mu)^2, \tag{2}$$

where $\mu$ is the mean value of $R_i^n$. In this way, after the application of a series of sliding calculations on the RSS values of link $L_i$, we can obtain a corresponding variance value array $V_i$. Owing to the different variance values on different links, it is not convenient to set an appropriate triggering threshold for different links. Hence, min-max normalization is applied to the series of variance values. The minimum variance value is transformed to a zero value, the maximum variance value is transformed to unity, and every other value is transformed to a decimal value between zero and one. The normalized variance sequence $V_i'$ is as follows,

$$V_i' = \frac{V_i - min(V_i)}{max(V_i) - min(V_i)}, \tag{3}$$

where $max(\cdot)$ and $min(\cdot)$ denote the calculation operations of the maximum and minimum values. It should be noted that, the maximum and minimum of the variance values would be updated continuously as the RSS measurements are collected.

The wireless link can then be triggered by the target if some value $V_i$ in the corresponding normalized variance sequence $V_i'$ is above the triggering detection threshold $V_{th}$. The set $J_t$ of the triggered links at time $t$ is defined as the set containing all wireless links $L_i$ that satisfy $V_i > V_{th}$, that is,

$$J_t = \{L_i | V_i > V_{th}, V_i \in V_i'\}, \tag{4}$$

where $V_{th}$ is determined by researchers based on the different tracking environments to obtain the best tracking performance.

Figures 2 and 3 show that the raw RSS data collected from wireless sensors and the normalized filtered variance data using a sliding window filter. It can be observed that the proposed triggered link detection method can effectively capture the triggering time of the target. The performance at different lengths of sliding window $c$ and triggering detection threshold $V_{th}$ are discussed in Section 3.

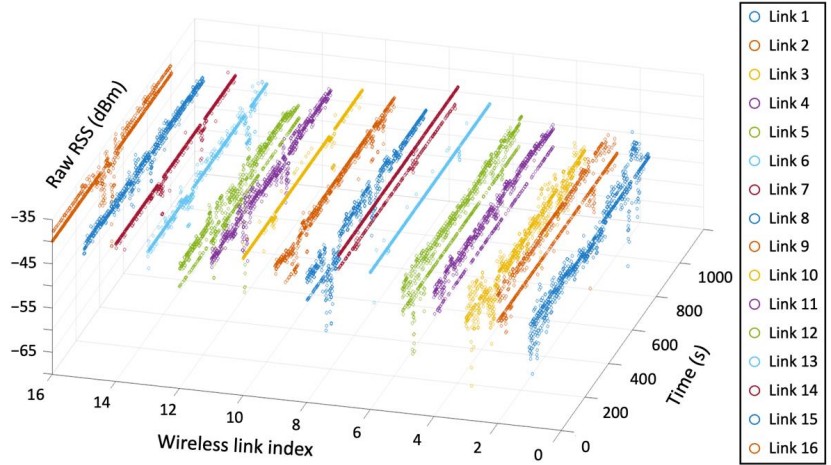

**Figure 2.** Raw received-signal strength (RSS) data of all the wireless links in the network.

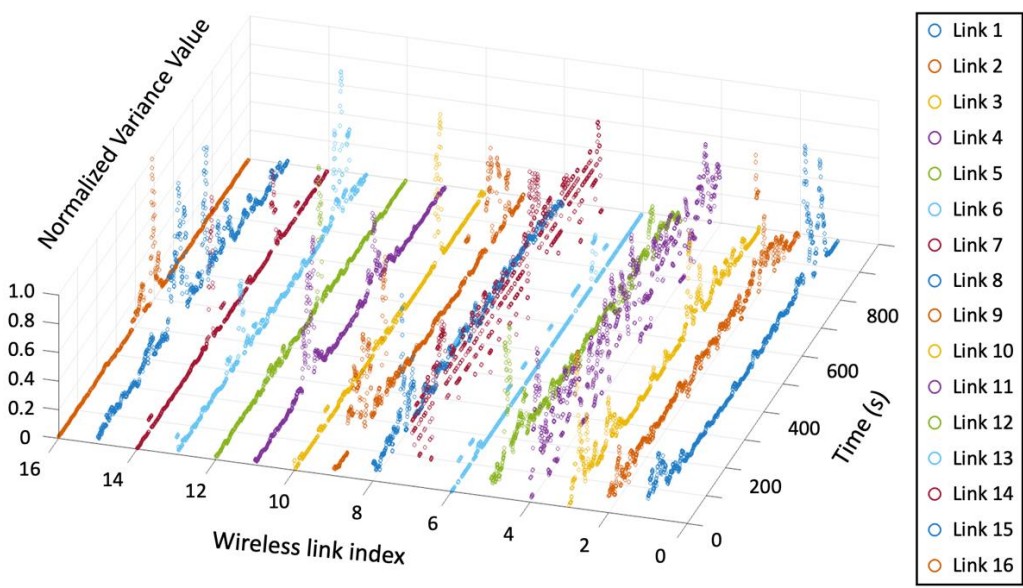

**Figure 3.** Plots of variances of the RSS data of all the wireless links in the network when the length of the sliding window *c* equals 15.

### 3.2. Intersection-Point Algorithm

The parametric forms of the links $L_j$ and $L_k$, from the node location $n_j^a$ to $n_j^b$, and from $n_k^a$ to $n_k^b$, respectively, are,

$$
\begin{aligned}
L_j &= \left[ \left( x_j^a, y_j^a \right), \left( x_j^b, y_j^b \right) \right], \\
L_k &= \left[ \left( x_k^a, y_k^a \right), \left( x_k^b, y_k^b \right) \right],
\end{aligned}
\tag{5}
$$

where $\left( x_j^a, y_j^a \right)$, $\left( x_j^b, y_j^b \right)$, $\left( x_k^a, y_k^a \right)$, and $\left( x_k^b, y_k^b \right)$ are the coordinates of points, $n_j^a$, $n_j^b$, $n_k^a$ and $n_k^b$, respectively. The direction vector of the link $L_j$ can be defined as,

$$
\begin{aligned}
g_j &= n_j^b - n_j^a \\
&= \left[ x_j^b - x_j^a, y_j^b - y_j^a \right],
\end{aligned}
\tag{6}
$$

Because the normal vector is perpendicular to the direction vector, the non-normalized normal vector of the link $L_j$ can be calculated as,

$$
h_j = \left[ y_j^b - y_j^a, x_j^a - x_j^b \right],
\tag{7}
$$

For two links with limited length, there are five types of relationships between them, as shown in Figure 4. The projection of the vector $n_j^a$ (we use $n_j^a$ as the notation of the vector from the origin of coordinates to the current node coordinates $n_j^a$ in the following description) onto $h_j$ can be expressed as,

$$
p_1 = n_j^a \cdot h_j,
\tag{8}
$$

where $(\cdot)$ is the dot product. In the same way, $p_2 = n_k^a \cdot h_k$. If the projection is from the vector $n_k^a$ to the normal vector $h_j$, it can be expressed as,

$$
p_{1a} = n_k^a \cdot h_j,
\tag{9}
$$

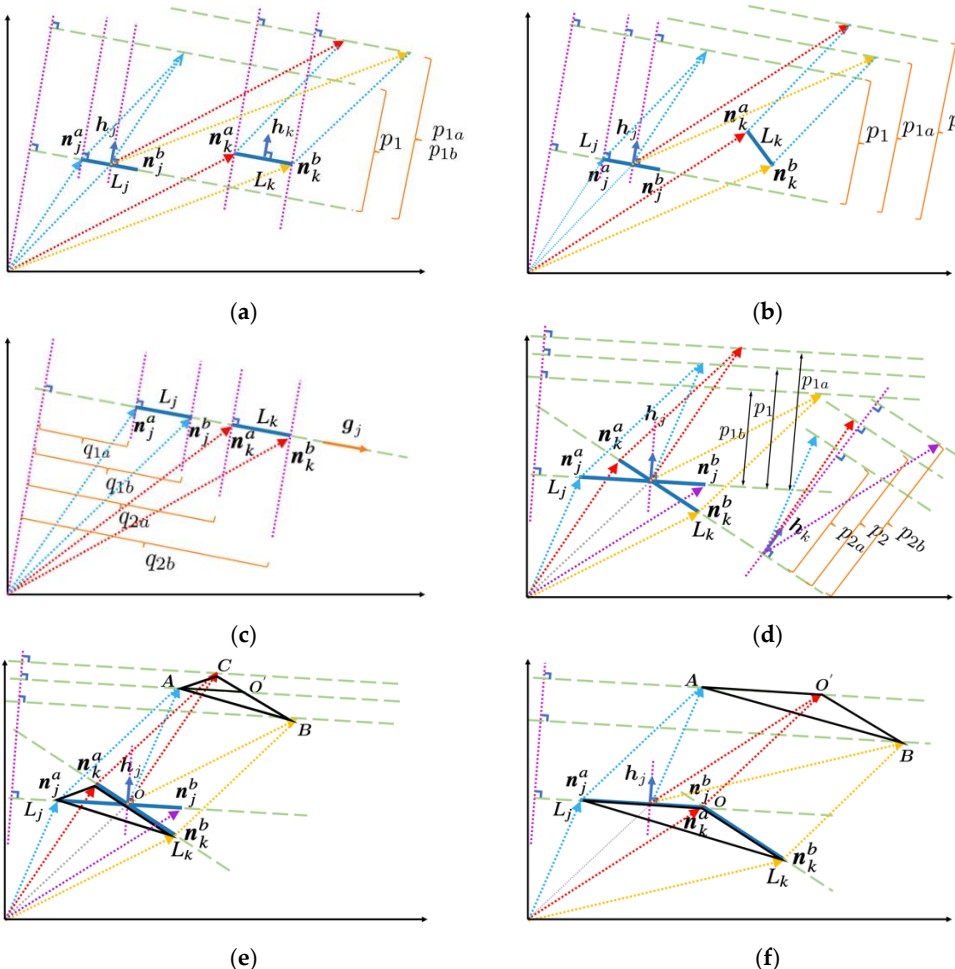

**Figure 4.** Illustrations of five possible location relationships of two links. (**a**) Case in which the two links are parallel with each other. (**b**) Case in which the two links are not parallel and do not intersect with each other. (**c**) Case in which the two links are collinear but not coincident. (**d**) Case in which the two links intersect but the intersection point does not exist at the endpoints. (**e**) A simplified sketch map of (**d**). (**f**) The two links intersect, but the intersection point exists at endpoints.

Similarly, the projection of the vector $n_k^b$ to a normal vector $h_j$ is expressed as $p_{1b} = n_k^b \cdot h_j$. Note that the subscript "1" is related to the normal vector $h_j$, and "2" is related to $h_k$. Therefore, the projection of vector $n_j^a$ and $n_j^b$ to the normal vector $h_k$ can be written as $p_{2a} = n_j^a \cdot h_k$ and $p_{2b} = n_j^b \cdot h_k$, respectively.

To describe the link intersection algorithm concisely, we also define the projection of the vector $n_j^a$ onto the direction vector $g_j$, which is expressed as,

$$q_{1a} = n_j^a \cdot g_j, \tag{10}$$

Similarly, $q_{1b} = n_j^b \cdot g_j$, the projection of vectors $n_k^a$ and $n_k^b$ onto the direction vector $g_k$ are $q_{2a} = n_k^a \cdot g_k$ and $q_{2b} = n_k^b \cdot g_k$, respectively.

As shown in Figure 4a, when the two links $L_j$ and $L_k$ are parallel to each other, the normal vectors $h_j$ and $h_k$ of the corresponding links are parallel. The projection of vectors $n_k^a$ and $n_k^b$ of the link $L_k$ on the normal vector $h_j$ are equal. The two values of $|p_{1a} - p_1|$ and $|p_{1b} - p_1|$ are equal and nonzero, which can be a criterion for determining the parallelism. In this case, there is no cross point between the two links. In Figure 4b, the two links are not parallel to each other, and do not intersect with each other. Therefore, the projection

values $|p_{1a} - p_1|$ and $|p_{1b} - p_1|$ are all nonzero and are not equal to each other. In this case, there is no intersection point between the two links. As a special case of the parallel orientation in Figure 4a, the collinearity of the two links is shown in Figure 4c. In this case, the three values of $p_1$, $p_{1a}$ and $p_{1b}$ are equal. To check whether there is an intersection point, four values $q_{1a}$, $q_{1b}$, $q_{2a}$ and $q_{2b}$ should be considered: the projections of vector $n_j^a$, $n_j^b$, $n_k^a$ and $n_k^b$ on the common direction vector $g_j$ or $g_k$, respectively. If the two links are collinear and there is no coincidence, as shown in Figure 4c, it should meet at least one of the two conditions that the product of $q_{1a} - q_{2a}$ and $q_{1b} - q_{2b}$ should be negative or approximately equal to zero, and the product of $q_{1b} - q_{2a}$ and $q_{1b} - q_{2b}$ should be negative or approximately equal to zero. Otherwise, the two links are collinear and overlap in part with each other. Because the case of two overlapping links is not a realistic scenario, it is omitted in Figure 4.

The most common case is shown in Figure 4d, where the two links intersect with each other. Because of the existence of the intersection point, the projection value $p_1$ should be between $p_{1a}$ and $p_{1b}$. Similarly, the projection value $p_2$ should be between $p_{2a}$ and $p_{2b}$. In other words, the product of $p_1 - p_{1a}$ and $p_1 - p_{1b}$ should be negative or approximately zero when the intersection point is infinitely close to any vertex of the corresponding link, as shown in Figure 4f. The projection $p_2$ should be calculated in a similar manner.

The method used to calculate the intersection point is shown in Figure 4e, which is a related to Figure 4d. The triangle $\Delta n_j^a n_k^a n_k^b$ is congruent with the triangle $\Delta ACB$, which is easily known. The ratio with $n_k^a O$ to $On_k^b$ can be calculated as,

$$\beta = \frac{\Gamma_{\Delta n_j^a n_k^a O}}{\Gamma_{\Delta n_j^a On_k^b}} = \frac{\Gamma_{\Delta ACO'}}{\Gamma_{\Delta AO'B}} = \frac{|p_1 - p_{1a}|}{|p_1 - p_{1b}|}, \tag{11}$$

where $\Gamma$ is the area of the triangle. Therefore, the intersection point $O$ can be calculated as

$$O = n_k^a + \left( n_k^b - n_k^a \right) * \beta, \tag{12}$$

The last case of the position relation of the two links is that some endpoint of the link is the intersection point, as shown in Figure 4f. In this case, the ratio $\beta = 0$, and the intersection point can be $n_k^a$. The intersection point algorithm is summarized in Algorithm 1.

---

**Algorithm 1** Intersection Point Algorithm

---

**Input**: The two links $L_j$ and $L_k$.
**Output**: The intersection points if it exists.
1. Calculate direction vector $g_j$ and $g_k$ using Equation (6).
2. Calculate normal vector $h_j$ and $h_k$ using Equation (7).
3. Calculate the projection of $p_1$, $p_{1a}$, $p_{1b}$, $p_2$, $p_{2a}$ and $p_{2b}$ using Equations (8) and (9).
4. **if** $|p_{1a} - p_1| = 0$ and $|p_{1b} - p_1| = 0$ **then**
5. 　　 $q_{1a} = n_j^a \cdot g_j$, $q_{1b} = n_j^b \cdot g_j$ 　　 $\triangleright$ *corr. to Equation (10)*
6. 　　 $q_{2a} = n_k^a \cdot g_k$, $q_{2b} = n_k^b \cdot g_k$
7. 　　 **if** $(q_{1a} - q_{2a})(q_{1b} - q_{2b}) \leq 0$ *or* $(q_{1b} - q_{2a})(q_{1b} - q_{2b}) \leq 0$ **then**
8. 　　　　 **return** $[-3, -3]$ $\triangleright$ *corr. to collinear and coincident*
9. 　　 **else**
10. 　　　　 **return** $[-2, -2]$ $\triangleright$ *corr. to Figure 4c*
11. 　　 **end**
12. **else if** $(p_{1a} - p_1)(p_{1b} - p_1) \leq 0$ *and* $(p_{2a} - p_2)(p_{2b} - p_2) \leq 0$ **then**
13. 　　 $\beta = \frac{|p_1 - p_{1a}|}{|p_1 - p_{1b}|}$
14. 　　 $cp = n_k^a + \left( n_k^b - n_k^a \right) * \beta$ 　　　　 $\triangleright$ *corr. to Equation (12)*
15. 　　 **return** $cp$ 　 $\triangleright$ *corr. to Figure 4d–f*
16. **else**
17. 　　 **return** $[-1, -1]$ $\triangleright$ *corr. to Figure 4a,b*
18. **end**

---

### 3.3. Build Segments of Links

Given the algorithm used to calculate the intersection point of links, we calculated the intersection points pairwise among all members of $L$. The set $Sp_j$ of link $L_j$ stores all the intersection points with other links. It should be noted that if an effective intersection point $cp$ between link $L_j$ and $L_k$ is calculated by excluding coinciding at the endpoints of links, it will be stored in the two sets $Sp_j$ and $Sp_k$ in the cases of the two corresponding links. Subsequently, the traversal loop is used to calculate and store the intersection points for all the links in the network. For the link $L_j$, all the intersection points in $Sp_j$ together with the the two endpoints $n_j^a$ and $n_j^b$ should be sorted by the $x$-axis coordinate in ascending order. The two connected points in the merged set $Sp_j$ form a new segment. All segments on the link $L_j$ are stored in the corresponding set $Seg_j$. A summary of the procedure for building the segment of links is presented in Algorithm 2.

---

**Algorithm 2** Calculate intersection point and segments of links

---

**Input**: The set of all the links $L$
**Output**: The set of all the links $L$ with segments information
1.    Initialize $Sp$ and $Seg$ for each link $L$ to the empty set.
2.    **for** all links $L_j$ in $L$ **do**
3.       **for** all links $L_k$ in $L$, $j \neq k$ **do**
4.       Solve the intersection point $cp_{jk}$ using Algorithm 1.
5.       **if** $cp_{jk} \notin \left\{ n_j^a, n_j^b \right\}$ **then**
6.         $Sp_j \cup \left\{ cp_{jk} \right\}$
7.         $Sp_k \cup \left\{ cp_{jk} \right\}$
8.       **end**
9.       **end**
10.    **end**
11.    **for** all links $L_j$ in $L$ do
12.       $Sp_j' = Sp_j \cup \left\{ n_j^a, n_j^b \right\}$
13.       Sort the points in set $Sp_j'$ by $x$-axis coordinate.
14.       **for** all links point $p_i$ in $Sp_j'$ **do**
15.       Build new segment $seg_i$ using $\{p_i, p_{i+1}\}$.
16.       $Seg_j \cup \{seg_i\}$
17.       **end**
18.    **end**

---

### 3.4. Compute the Segment Midpoint

The general equation of the link $L_j$ from node $n_j^a$ to $n_j^b$ is $A_j x + B_j y + C_j = 0$, where $A_j = y_j^b - y_j^a$, $B_j = x_j^a - x_j^b$ and $C_j = x_j^b * y_j^a - x_j^a * y_j^b$. The slope $k_j$ and intercept $b_j$ of the link $L_j$ were $k_j = -\left( A_j / B_j \right)$, $b_j = -\left( C_j / B_j \right)$, if $B_j \neq 0$. Draw a perpendicular to the link $L_j$ through the previous location estimate $\hat{x}_{t-1}$ with coordinates $(x_{t-1}, y_{t-1})$ as shown in Figure 5a. The slope and intercept of the perpendicular can be estimated as,

$$k_\perp = -\frac{1}{h_j}, \tag{13}$$

$$b_\perp = y_{t-1} - k_\perp * x_{t-1}, \tag{14}$$

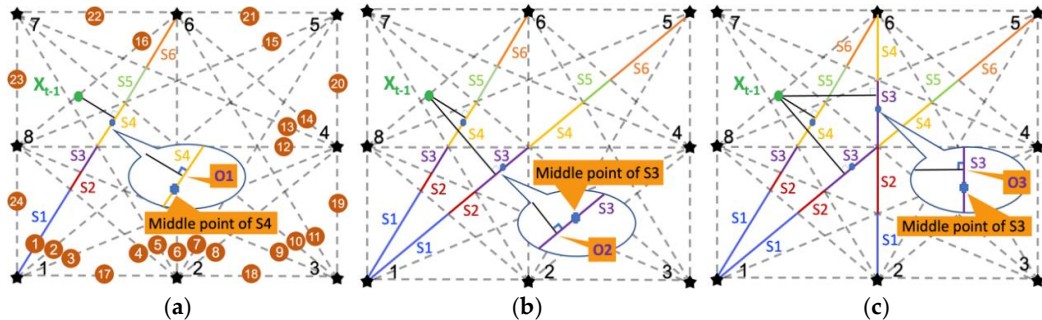

**Figure 5.** This is a figure. Schemes follow the same formatting. Illustration of the proposed geometric midpoint (GM) algorithm. Target estimation when (**a**) the quantity of triggered link is equal to one. (**b**) The quantity of triggered links is equal to two. (**c**) The quantity of triggered links is equal to three.

The intersection point $p$ ($O1$ in Figure 5a) with coordinates $(x_p, y_p)$ can be solved by using the point-slope equations,

$$y_p = k_\perp * x_p + b_\perp, \tag{15}$$

$$y_p = k_j * x_p + b_j, \tag{16}$$

For each segment in set $Seg_j$ on link $L_j$, find the segment $seg_j^*$ in which the middle point has the shortest distance with the intersection point $p$,

$$seg_j^* = \operatorname*{argmin}_{seg_i} d(mid(seg_i), p), \tag{17}$$

where $d(\cdot)$ represents the Euclidean distance operation, the calculation $mid(\cdot)$ for the middle point of the segment $seg_i$ with two endpoints $\left[p_i^a, p_i^b\right]$ is defined as,

$$mid(seg_i) = \frac{1}{2}\left[\left(x_{p_i^a} + x_{p_i^b}\right), \left(y_{p_i^a} + y_{p_i^b}\right)\right], \tag{18}$$

where $\left(x_{p_i^a}, y_{p_i^a}\right)$ and $\left(x_{p_i^b}, y_{p_i^b}\right)$ are the coordinates of the endpoints $p_i^a$ and $p_i^b$, respectively.

The middle point $q^j$ of the segment (such as the middle point of segment $S4$ in Figure 5a) on the link $L_j$ nearest to the previous location estimate $\hat{x}_{t-1}$ is given by,

$$q^j = mid\left(seg_j^*\right), \tag{19}$$

All the middle points of the nearest segment of links in $J_t$ are stored in the set $Q_t = \{q_t^1, q_t^2, \ldots, q_t^K\}$, where $K = |J_t|$ denotes the cardinality of the set $J_t$. For example, the set $Q_t$ includes the middle point of segment $S4$ on link 1 in Figure 5a, the middle point of segment $S3$ on link 2 in Figure 5b, and the middle point of segment $S3$ on link 6 in Figure 5c, if links 1, 2, and 6 are triggered.

### 3.5. Distance-Based Weights

To decrease the tracking error caused by the multipath effect, weights which are dependent on the distance from the middle point of the segment to the previous location estimate $\hat{x}_{t-1}$ are proposed. Distance-based weight $w_d^i$ for the middle point $q_t^i \in Q_t$ is computed as,

$$w_d^i = \frac{1}{||\hat{x}_{t-1} - q_t^i||}, \tag{20}$$

The middle points on segments which are closer to the previous location estimate are assigned higher weights compared with the middle points which are farther away. The set of distance-based weights $w_d^i$ for all middle points $q_t^i \in Q_t$ is defined as,

$$W^d = \left\{ w_d^i \middle| q_t^i \in Q_t \right\},$$

(21)

It is noted that when the first location estimate is calculated, that is, $t = 1$, all the weights $w_d^i$ are set to one.

### 3.6. Generation of Location Estimates

Given the set of middle points $Q_t$ and the distance-based weight $W^d$, the location estimate $\hat{x}_t$ can be calculated as the weighted mean of all the middle points in $Q_t$,

$$\hat{x}_t = \sum_{u=1}^{|J_t|} \frac{w_d^u}{\sum_{v=1}^{|J_t|} w_d^v} \times q_t^u,$$

(22)

where $|J_t|$ denotes the cardinality of the set $J_t$, and $q_t^u$ and $w_d^u$ represent the middle points of the segment on the $u$-th link $L_u$ on $J_t$ and the corresponding weight value, respectively. The GM algorithm is summarized in Algorithm 3, and an illustration is shown in Figure 5.

---

**Algorithm 3** Geometric midpoint algorithm for DFL

---

1. Calculate the segments of among all the links $L$ in the network using Algorithm 2.
2. After the initialization of network, collect the raw RSS data and store in $R$.
3. **for** $t \geq 1$ **do**
4.   Apply sliding window filter using Equations (1)–(3).
5.   Determine the triggered links $J_t$ using Equation (4).
6.   Generate the set of middle points $Q_t$ using Equations (13)–(19).
7.   Generate the set of distance-based weights $W^d$ using Equations (20) and (21).
8.   Calculate the location estimate $\hat{x}_t$ using Equation (22).
9. **end**

---

## 4. Performance Analysis and Evaluation

In this section, we evaluate the performance of the proposed GM algorithm based on the DFL system using simulation analyses and experimental evaluations.

Before the experiment, we performed a group of simulations to test the ideal performance of the proposed algorithm. We designed a typical target path, as shown in Figure 6a. We used a circle in two dimensions to represent the target. The vertical distance from the center of the circle to the link represents whether the target can trigger the link. If the vertical distance is lower than a threshold value, the link can be regarded as being triggered. In other words, this process does not include the simulation for wireless signals, but after the link triggering information is determined by the pure geometric distance between target (circle) and links (lines), is then to feed the triggered link information to the GM scheme and acquire the final target estimate. In this way, a dynamic simulation process is performed, and the visualized tracking performance of the simulation and mean tracking errors are shown in the Figure 7a,b.

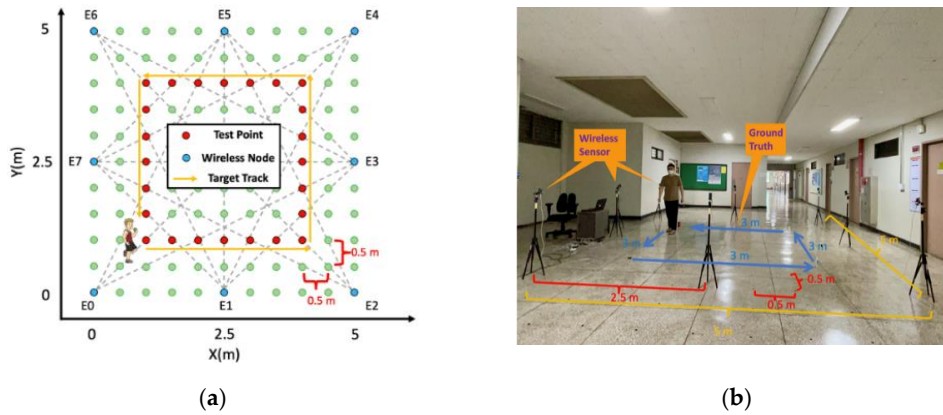

(a)

(b)

**Figure 6.** (**a**) Experimental setup. (**b**) Photograph of the experiment setup with the target traversing the predefined path.

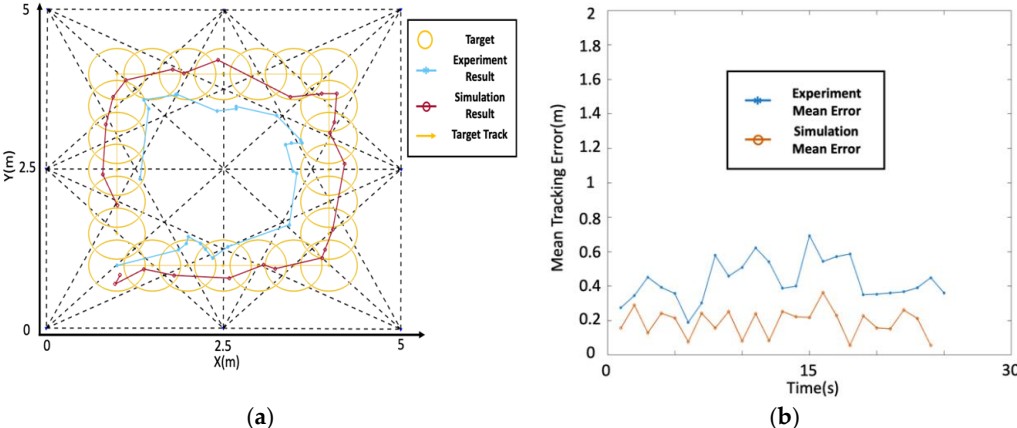

(a)

(b)

**Figure 7.** (**a**) Visualized tracking performance of the proposed GM algorithm. (**b**) Mean tracking errors between the predefined path and the estimated paths.

The average error of the estimated target location is defined as,

$$\varepsilon = \frac{1}{K_s} \sum_{k=1}^{K_s} \sqrt{(x_k - \hat{x}_k)^2 + (y_k - \hat{y}_k)^2},$$
(23)

where $K_s$ is the total number of samples, $x_k$ and $y_k$ are the actual target location coordinates of the *x*-axis and *y*-axis, respectively, and $\hat{x}_k$ and $\hat{y}_k$ are the estimated *x* and *y* coordinates at sample time *k*.

*4.1. Experimental Setup*

To evaluate the performance of our proposed GM algorithm in an actual indoor physical environment, we performed multiple experiments in different days to validate the practicality and robustness of the proposed scheme. The experiments are performed on a wireless peer-to-peer sensor network containing nine nodes. In the experiments, the target moved according to the predefined path. Eight of the nine nodes are distributed on both sides of the 5 m × 5 m square area, and each node is placed 2.5 m apart on a side. The wireless nodes operated on a 2.4 GHz industrial, scientific, and medical (ISM) band based on the IEEE 802.15.4 standard and were placed on tripods approximately 1.3 m above the ground. That is because the main localization target is human beings with height from 1.6 m to 1.8 m. Nodes 1–8 are normal nodes and broadcast signals sequentially at fixed time intervals. The node that broadcasted signals worked as a transmitter, and the remaining seven nodes worked as receivers. The node 9 was the central node used to coordinate

the operation of the network, such as sending the transmit command to the transmitters. The transmitters broadcasted signals to the receivers sequentially every 0.1 s to prevent transmission collisions. Once the receiver received the signal from the transmitter, it extracted the RSS and source address ID information from the frame, and fed the data together with their ID information to a laptop computer via a universal asynchronous receiver/transmitter port. An illustration and photograph of the experimental setup are shown in Figure 6a,b.

The default parameters are summarized as follows: the speed of the target is approximately 0.5 m/s, the length of the sliding window filter $c = 16$, and the threshold value for triggering detection $V_{th} = 0.4$. The performance of the proposed GM algorithm is compared with that of the state-of-the-art GF algorithm. The algorithms were coded in MATLAB (version R2021a, MathWorks, Natick, MA, USA), and ran on a 3.40-GHz desktop computer.

*4.2. Experimental Results and Analysis*

The results of the experiment using the proposed GM algorithm are shown in Figure 7a, and the mean tracking errors are shown in Figure 7b. The experiment result in Figure 7b is realized under the parameter the length of sliding window filter $c = 16$, the triggering threshold $V_{th} = 0.4$. As shown, our proposed GM algorithm can track moving targets with small errors even in low-density wireless networks.

The performance of the proposed algorithm was evaluated subject to the two main parameters sliding window length $c$ and triggering detection threshold $V_{th}$, which are related to the triggering of wireless links; relevant results are shown in Figure 8. All the cases for various triggering detection threshold (from 0.1 to 0.9) and various length of sliding window (from 5 to 39) are included to calculate the RMSE, and the vertical lines show the range of the RMSE results. As shown in Figure 8a, if the length of sliding window $c$ is too short, the range of error is too large, and the system cannot track the target with stable accuracy. The reason is that a very small $c$ value cannot efficiently capture the fluctuation caused by the target. As the length of the sliding window $c$ increases, the tracking errors also increase. This is because a larger $c$ will increase the computation of filtering by the sliding window and decrease the ability to detect the fluctuation of wireless signals. Hence, the appropriate length of the sliding window $c$ was between 16 and 20.

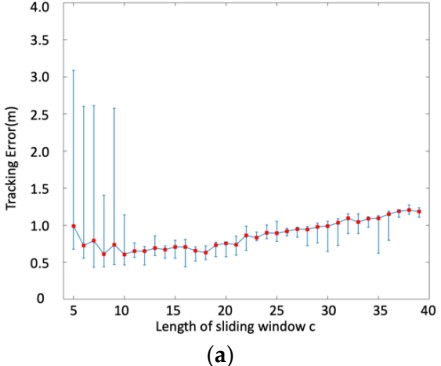 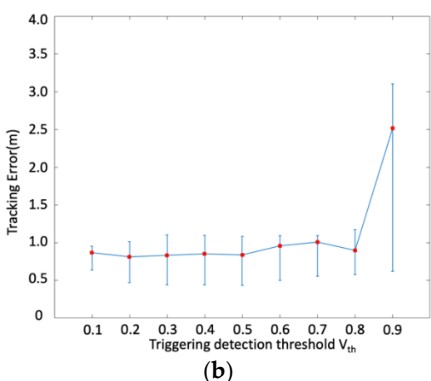

(**a**)　　　　　　　　　　　　　　(**b**)

**Figure 8.** Tracking errors as functions of the length of the sliding window and triggering detection threshold for the proposed GM scheme (**a**) length of sliding window $c$ and (**b**) triggering detection threshold $V_{th}$. The central marks indicate the medians of tracking errors. The lower and upper bars indicate the 0th and 75th percentiles, respectively.

Figure 8b shows that the mean tracking errors increase as the triggering detection threshold $V_{th}$ increases. A very large $V_{th}$ causes the system to ignore some real but small fluctuations caused by the target, and decreases the accuracy of the system. If the threshold $V_{th}$ is too small, it causes false triggering. Because this threshold is applied to the normal-

ized variance signals, small values can still ensure that the system maintains good tracking accuracy. The appropriate threshold value for this experiment was 0.4.

To demonstrate the advantages of the proposed GM algorithm in a low-density wireless network, we used the same dataset to test the performance of the conventional state-of-the-art device-free localization scheme (the GF scheme) [18], which can achieve high-tracking accuracy in high-density wireless networks. The results are presented in Figure 9. All the cases for various triggering detection threshold (from 0.1 to 0.9) and various length of sliding window (from 5 to 39) are included to calculate the RMSE, and the vertical lines show the range of the RMSE results. As shown in Figure 9a, the errors associated with the GF scheme in the $3 \times 3$ wireless network are much higher than the mean error of the proposed GM scheme, which is approximately 3 m. The triggering threshold value (which is below 0.3) can realize relatively high accuracy (approximately equal to 0.8 m) as shown in Figure 9b. In GF scheme, the researchers performed the experiments with $5 \times 5$ wireless network (network density is about $1.25\,\text{link}/\text{m}^2$) in outdoor environment, which realized high tracking accuracy. Generally speaking, it is necessary to build high density wireless network to realize high tracking accuracy. The accuracy of wireless network-based tracking system in outdoor is higher than in indoor environment, due to the severe multipath effect in indoor environment. Our proposed scheme is designed for low density network (about $0.64\,\text{link}/\text{m}^2$) and finally realized relatively higher tracking accuracy than the conventional GF scheme when facing the two challenges–low network density and indoor environment.

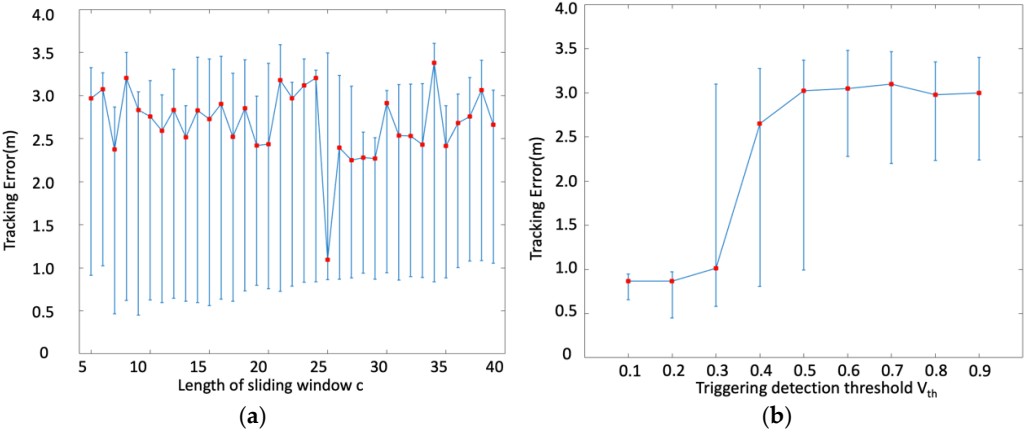

**Figure 9.** Tracking errors as functions of the length of the sliding window and triggering detection threshold for the conventional GF scheme (**a**) length of sliding window *c*, and (**b**) triggering detection threshold $V_{th}$. The central marks indicate the medians of tracking errors. The lower and upper bars indicate the 0th and 75th percentiles, respectively.

Table 1 lists the statistical minimum, median and mean values for the tracking root-mean-squared errors (RMSE) subject to Figures 8a and 9a. From Table 1, we can see that even the minimum value is similar for the proposed GM scheme and the conventional GF scheme. But the proposed GM scheme outperform GF scheme in median and mean values, with 68.42% and 67.66% increase, respectively, which shows the GM scheme has more advantages in low density wireless networks.

**Table 1.** Statistical characteristics of root-mean-square-errors (RMSE).

| Algorithm | Min (m) | Median (m) | Mean (m) |
|---|---|---|---|
| RSS based GM | 0.4367 | 0.8603 | 0.8679 |
| RSS based GF | 0.4516 | 2.7241 | 2.6837 |

Finally, we list some problems we meet during our experiment that we set to finish in future. Limited by the hardware performance of our wireless module, users could not modify the sampling rate, which caused the system cannot collected enough RSS data in short time. However, the length of the sliding window is related to the input RSS data. If the length of the sliding window is set to a very large value, the system will keep waiting until enough RSS data is fed by serial port, which directly causes the decrease of tracking performance in real-time localization application. What's more, if the sampling rate of the wireless modules can increase, the system delay will significantly decrease and get good tracking accuracy.

## 5. Conclusions

In this study, we firstly proposed an effective wireless link detection method that can detect the fluctuations caused by targets and then proposed a geometrical midpoint algorithm that can track a target effectively in a low-density WSN. In this scheme, we used the middle points of segments that were formed by the intersections among links as the possible positions of the target. The target estimate was combined with all the distance-based weighted midpoints of the related segments on the triggered links. This proposed scheme realized the average tracking accuracy about 0.8 m level, while the conventional GF scheme is about 2.6 m level when the size of monitored area is 25 $m^2$. This method did not require complex computations, such as particle filter-based or grid-based algorithms, which made it suitable for use in resource-limited scenarios. Our future work includes: perform more groups of experiment under different conditions to test the practicality and robustness of the proposed scheme; calculate the computational complexity of the proposed scheme and compare with conventional methods; test the performance of the proposed method in wider monitored area and more nosier environment; test the possibility of the proposed scheme for tracking multiple targets with acceptable accuracy; and expand the proposed scheme to make it to be applied in 3-dimensional environment.

**Author Contributions:** Conceptualization, methodology, and writing—original draft preparation, C.S.; validation and data curation, B.Z. and S.Y.; writing—review and editing and supervision, Y.K. All authors have read and agreed to the published version of the manuscript.

**Funding:** This work was supported by the National Research Foundation of Korea (NRF) grant funded by the Korea Government (MSIT) (NRF-2021R1F1A1049509). The present research has been conducted by the Research Grant of Kwangwoon University in 2021. This work was supported by the Opening Foundation of Key Laboratory of Dynamic Cognitive System of Electromagnetic Spectrum Space, Ministry of Industry and Information Technology (KF20202104).

**Conflicts of Interest:** The authors declare no conflict of interest.

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
