# Peer review of "Geometric Midpoint Algorithm for Device-Free Localization in Low-Density Wireless Sensor Networks"

_electronics, doi:10.3390/electronics10232924_

Round 1

Reviewer 1 Report

The introduction provides sufficient background and a neat opening to the project, highlighting relevant papers which work on other methods of localisation. More references could have been used to back up points such as cameras and infrared sensors only being used for restricted tasks.

The research design is appropriate in the way tags are detected and the theory of using the intersection point to calculate the tags’ location. However, having each node in the WSN at a height of 1.3m is not realistic in terms of what environment the paper has suggested the localisation technique could be used in.

As well as the height of the nodes, possible extensions of the experiment could show how certain obstacles alter results. Results of extending the range of nodes were simulated although it may have been useful to see this done physically too.

The methods are adequately described. It is clear to see the intention of the work suggested. Each phase is appropriately described, from detecting links in the network to applying the GM algorithm.

The results are clearly presented. All figures used are clear, they give a good indication of how the experiment was set up, and how the results compared with the simulation technique and the physical experiment.

The conclusion summarised the work, however it does not mention certain results of the experiment, for example, the accuracy level being 0.8m.

Reviewer 2 Report

In general, the paper is well written, and it sounds scientifically correct. However, my major concern regards the lack of background and related works to support the claims of originality. The authors must insert more related works (recent ones) and a table comparing the proposal of this article with correlated ones. The other major reviews are:

The 3rd and 4th claims are not contributions. Please, correct it.

Figures 1 and 2 are exhibited before they are called in the text. This generates confusion. Please, fix it.

Insert a picture from the indoor physical scenario detailing the experiment setup up, or improve Figure 6 (b). For instance, showing the ground truth

Insert more quantitative results. Besides, How is the performance of your technique under different conditions? What can be inferred of practicality and robustness?

Insert a discussion comparing the obtained results with the literature.

The conclusion section should give ideas for future works.

Reviewer 3 Report

In the presented paper authors proposed the geometric midpoint algorithm for the device-free localization. The solution is designed to operate in indoor environment.

The paper presents an interesting concept, but some significant drawbacks can be pointed.

  1. The Introduction section provide basic information, but it is too simplified. Authors must split it to Introduction and Related Work sections. Moreover only 11 references were pointed in the paper, this is not enough. Author should also point solutions basing on the channel impulse response.
  2. The structure of the manuscript is not correct. All the figures must be placed after they are mentioned in the text. When they occur first, the reader does not know what they present – like Fig. 1,2,3 and others in the manuscript.
  3. In the Fig. 2 the axil label point RSSI, not RSS. It must be unified in the paper.
  4. The description of the algorithm, Section 2, is not clear for the reader. In general it is extremely hard to understand the principle of operation of the proposed method. The included pseudocodes are helpful, but still the reader must read some sentences a few times to understand the authors’ point of view. This part should be extensively reviewed.
  5. Authors mentioned that the transmission period equals 0.1 ms, but how long was the transmission? The RSS value for one link was determined wit 0.8 ms period? It seems that very few values of RSS are collected during the movement of the person (the presented scenario).
  6. The error of the position estimates seems to be influenced by a “bias” – the values of position estimates are concentrated toward the center point.
  7. In the conclusions author mentioned about the comparison of the computational complexity (regarding e.g. the particle filtration), it was not described in the manuscript.
  8. The values of the RSS presented in the figure 2 are high (about -40 to -50 dBm) – the quality of the received signal was good. How the proposed method will work when the received signal strength will be lower, and the channel will be more noisy?

The English language check should be done, the mistakes can be found in each paragraph.

Reviewer 4 Report

The paper deals with a novel solution for device-free positioning. 

  1. A more detailed overview of the state of the art solutions should be provided. The authors compare the proposed GM method to the GF method, which is not explained in the paper.
  2. The results provided in the paper consist of experiential results and simulation results. However, there is no information about the simulation scenario, settings of the simulation model etc. To understand the results it is needed to know how fluctuations of the signal caused by the moving object were simulated. 
  3. In the experiment, the size of the sliding window was set to 16. If the sliding window is 16 how much time does it take to estimate position? Single RSS measurement from all nodes may take 0.9s (9 nodes x 0.1s sequential transmission).
  4. Results presented in Table 1 are not in line with the results presented in figures 8 and 9. From the figures, it seems that the achieved error is different from the errors presented in the table. 
  5. Is the algorithm able to track multiple persons moving in the area and clearly distinguish between them?
  6. In equation 3, it is not quite clear how are max and min values are estimated. Is it only based on sampled data from the current time window?
  7. The discussion of the presented results should be more detailed.

Reviewer 5 Report

A interesting method to obtain the position using node links. The paper is well organized and given details are sufficient so that non-experts have no issues to understand the method. However i have one point: is it possible to use probabilistic filter to determine the target location instead of conducting presented steps? E.g to model the information of target located on particular link as a close-form equation and use Kalman filter to estimate the target's location?? In such way the multiple steps in the paper can be avoided. 

Round 2

Reviewer 1 Report

Authors have successfully addressed all my comments and concerns. 

Author Response

We appreciate for the reviewer’s positive recognition of our research work. We will do our best to proceed our future works. 

Reviewer 2 Report

The authors improved the manuscript and performed the suggestions. However, the authors should improve the paper's references by adding more recent works (2020 and 2021), and improving the Related Works Section - ten papers is a good number. At the end of this section, the authors must give the advantages of their proposition compared with the mentioned works. The discussion and comparison of what is being proposed in the literature are crucial for research papers. It gives the reader the big picture of what novelty is.

Reviewer 3 Report

The changes introduced by the authors are sufficient to publish the article in the current form.

Author Response

(The authors gave the same response as above.)

Reviewer 4 Report

The manuscript was improved according to the comments from the first round of review. 

However, there are still some unresolved issues, that should be addressed. 

"... max and min RSS values are calculated based on a period sample time rather than only within current time window." 

It is not clear what authors refer to as "period sample time" this should be defined clearly, since it has significant impact on function of the proposed solution. 

It is not clear what size of sliding window and what triggering detection threshold was used to achieve experimental results presented in figure 7b. 

What triggering detection threshold was used to obtain results presented in figures 8a and 9a?

What was the length of sliding window in figures 8b and 9b?

Why did authors select those specific values of  sliding window length and triggering detection threshold?

It would be better to show 3D figure (i.e. mean error as function of sliding window length and triggering detection threshold) to see impact of both and then present results for selected values. 

Round 3

Reviewer 2 Report

The authors performed all suggestions raised in the first review. I recommend checking the grammar spelling.